# Targeting EEG/LFP Synchrony with Neural Nets

Yitong Li[1], Michael Murias[2], Samantha Major[2], Geraldine Dawson[2], Kafui Dzirasa[2],
Lawrence Carin[1] and David E. Carlson[3,4]

[1]Department of Electrical and Computer Engineering, Duke University
[2]Departments of Psychiatry and Behavioral Sciences, Duke University
[3]Department of Civil and Environmental Engineering, Duke University
[4]Department of Biostatistics and Bioinformatics, Duke University
{yitong.li,michael.murias,samantha.major,geraldine.dawson,
kafui.dzirasa,lcarin,david.carlson}@duke.edu

## Abstract

We consider the analysis of Electroencephalography (EEG) and Local Field Potential (LFP) datasets, which are "big" in terms of the size of recorded data but rarely have sufficient labels required to train complex models (*e.g.*, *conventional* deep learning methods). Furthermore, in many scientific applications, the goal is to be able to understand the underlying features related to the classification, which prohibits the blind application of deep networks. This motivates the development of a new model based on *parameterized* convolutional filters guided by previous neuroscience research; the filters learn relevant frequency bands while targeting synchrony, which are frequency-specific power and phase correlations between electrodes. This results in a highly expressive convolutional neural network with only a few hundred parameters, applicable to smaller datasets. The proposed approach is demonstrated to yield competitive (often state-of-the-art) predictive performance during our empirical tests while yielding *interpretable* features. Furthermore, a Gaussian process adapter is developed to combine analysis over distinct electrode layouts, allowing the joint processing of multiple datasets to address overfitting and improve generalizability. Finally, it is demonstrated that the proposed framework effectively tracks neural dynamics on children in a clinical trial on Autism Spectrum Disorder.

## 1 Introduction

There is significant current research on methods for Electroencephalography (EEG) and Local Field Potential (LFP) data in a variety of applications, such as Brain-Machine Interfaces (BCIs) [21], seizure detection [24, 26], and fundamental research in fields such as psychiatry [11]. The wide variety of applications has resulted in many analysis approaches and packages, such as Independent Component Analysis in EEGLAB [8], and a variety of standard machine learning approaches in FieldTrip [22]. While in many applications prediction is key, such as for BCIs [18, 19], in applications such as emotion processing and psychiatric disorders, clinicians are ultimately interested in the dynamics of underlying neural signals to help elucidate understanding and design future experiments. This goal necessitates development of *interpretable* models, such that a practitioner may *understand* the features and their relationships to outcomes. Thus, the focus here is on developing an interpretable and predictive approach to understanding spontaneous neural activity.

A popular feature in these analyses is based on spectral coherence, where a specific frequency band is compared between pairwise channels, to analyze both amplitude and phase coherence. When two regions have a high power (amplitude) coherence in a spectral band, it implies that these areas are

coordinating in a functional network to perform a task [3]. Spectral coherence has been previously used to design classification algorithms on EEG [20] and LFP [30] data. Furthermore, these features have underlying neural relationships that can be used to design causal studies using neurostimulation [11]. However, fully pairwise approaches face significant challenges with limited data because of the proliferation of features when considering pairwise properties. Recent approaches to this problem include first partitioning the data to spatial areas and considering only broad relationships between spatial regions [33], or enforcing a low-rank structure on the pairwise relationships [30].

To analyze both LFP and EEG data, we follow [30] to focus on low-rank properties; however, this previous approach focused on a Gaussian process implementation for LFPs, that does not scale to the greater number of electrodes used in EEG. We therefore develop a new framework whereby the low-rank spectral patterns are approximated by *parameterized* linear projections, with the parametrization guided by neuroscience insights from [30]. Critically, these linear projections can be included in a convolutional neural network (CNN) architecture to facilitate end-to-end learning with *interpretable* convolutional filters and fast test-time performance. In addition to being interpretable, the parameterization dramatically reduces the total number of parameters to fit, yielding a CNN with only hundreds of parameters. By comparison, conventional deep models require learning millions of parameters. Even special-purpose networks such as EEGNet [15], a recently proposed CNN model for EEG data, still require learning thousands of parameters.

The parameterized convolutional layer in the proposed model is followed by max-pooling, a single fully-connected layer, and a cross-entropy classification loss; this leads to a clear relationship between the proposed targeted features and outcomes. When presenting the model, interpretation of the filters and the classification algorithms are discussed in detail. We also discuss how deeper structures can be developed on top of this approach. We demonstrate in the experiments that the proposed framework mitigates overfitting and yields improved predictive performance on several publicly available datasets.

In addition to developing a new neuroscience-motivated parametric CNN, there are several other contributions of this manuscript. First, a Gaussian Process (GP) adapter [16] within the proposed framework is developed. The idea is that the input electrodes are first mapped to pseudo-inputs by using a GP, which allows straightforward handling of missing (dropped or otherwise noise-corrupted) electrodes common in real datasets. In addition, this allows the same convolutional neural network to be applied to datasets recorded on distinct electrode layouts. By combining data sources, the result can better generalize to a population, which we demonstrate in the results by combining two datasets based on emotion recognition. We also developed an autoencoder version of the network to address overfitting concerns that are relevant when the total amount of *labeled* data is limited, while also improving model generalizability. The autoencoder can lead to minor improvements in performance, which is included in the Supplementary Material.

## 2 Basic Model Setup: Parametric CNN

The following notation is employed: scalars are lowercase italicized letters, e.g. $x$, vectors are bolded lowercase letters, e.g. $\boldsymbol{x}$, and matrices are bolded uppercase letters, e.g. $\boldsymbol{X}$. The convolution operator is denoted $*$, and $\jmath = \sqrt{-1}$. $\otimes$ denotes the Kronecker product. $\odot$ denotes an element-wise product.

The input data are $\boldsymbol{X}_i \in \mathbb{R}^{C \times T}$, where $C$ is the number of simultaneously recorded electrodes/channels, and $T$ is given by the sampling rate and time length; $i = 1, \ldots, N$, where $N$ is the total number of trials. The data can also be represented as $\boldsymbol{X}_i = [\boldsymbol{x}_{i1}, \cdots, \boldsymbol{x}_{iC}]^{\mathsf{T}}$, where $\boldsymbol{x}_{ic} \in \mathbb{R}^T$ is the data restricted to the $c$th channel. The associated labels are denoted $y_i$, which is an integer corresponding to a label. The trial index $i$ is added only when necessary for clarity.

An example signal is presented in Figure 1 (Left). The data are often windowed, the $i$th of which yields $\boldsymbol{X}_i$ and the associated label $y_i$. Clear identification of phase and power relationships among channels motivates the development of a structured neural network model for which the convolutional filters target this synchrony, or frequency-specific power and phase correlations.

### 2.1 SyncNet

Inspired both by the success of deep learning and spectral coherence as a predictive feature [12, 30], a CNN is developed to target these properties. The proposed model, termed SyncNet, performs a structured 1D convolution to jointly model the power, frequency and phase relationships between channels.

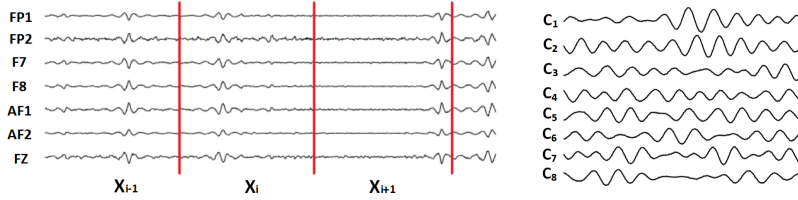

Figure 1: (Left) Visualization of EEG dataset on 8 electrodes split into windows. The markers (e.g., "FP1") denote electrode names, which have corresponding spatial locations. (Right) 8 channels of synthetic data. Refer to Section 2.2 for more detail.

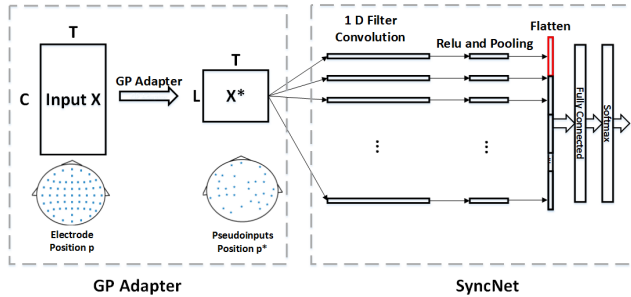

Figure 2: SyncNet follows a convolutional neural network structure. The right side is the SyncNet (Section 2.1), which is parameterized to target relevant quantities. The left side is the GP adapter, which aims at unifying different electrode layout and reducing overfitting (Section 3).

This goal is achieved by using *parameterized* 1-dimensional convolutional filters. Specifically, the $k$th of $K$ filters for channel $c$ is

$$f_c^{(k)}(\tau) = b_c^{(k)} \cos(\omega^{(k)}\tau + \phi_c^{(k)}) \exp(-\beta^{(k)}\tau^2). \qquad (1)$$

The frequency $\omega^{(k)} \in \mathbb{R}^+$ and decay $\beta^{(k)} \in \mathbb{R}^+$ parameters are shared across channels, and they define the real part of a (scaled) Morlet wavelet[1]. These two parameters define the spectral properties targeted by the $k$th filter, where $\omega^{(k)}$ controls the center of the frequency spectrum and $\beta^{(k)}$ controls the frequency-time precision trade-off. The amplitude $b_c^{(k)} \in \mathbb{R}^+$ and phase shift $\phi_c^{(k)} \in [0, 2\pi]$ are channel-specific. Thus, the convolutional filter in each channel will be a discretized version of a scaled and rotated Morlet wavelet. By parameterizing the model in this way, all channels are targeted collectively. The form in (1) is motivated by the work in [30], but the resulting model we develop is far more computationally efficient. A fuller discussion of the motivation for (1) is detailed in Section 2.2.

For practical reasons, the filters are restricted to have finite length $N_\tau$, and each time step $\tau$ takes an integer value from $\left[-\frac{N_\tau}{2}, \frac{N_\tau}{2} - 1\right]$ when $N_\tau$ is even and from $\left[-\frac{N_\tau-1}{2}, \frac{N_\tau-1}{2}\right]$ when $N_\tau$ is odd. For typical learned $\beta^{(k)}$'s, the convolutional filter vanishes by the edges of the window. Succinctly, the output of the $k$ convolutional filter bank is given by $\boldsymbol{h}^{(k)} = \sum_{c=1}^{C} f_c^{(k)}(\boldsymbol{\tau}) * \boldsymbol{x}_c$.

The simplest form of SyncNet contains only one convolution layer, as in Figure 2. The output from each filter bank $\boldsymbol{h}^{(k)}$ is passed through a Rectified Linear Unit (ReLU), followed by max pooling over the entire window, to return $\tilde{h}^{(k)}$ for each filter. The filter outputs $\tilde{h}^{(k)}$ for $k = 1, \ldots, K$ are concatenated and used as input to a softmax classifier with the cross-entropy loss to predict $\hat{y}$. Because of the temporal and spatial redundancies in EEG, dropout is instituted at the channel level, with

$$dropout(\boldsymbol{x}_c) = \begin{cases} \boldsymbol{x}_c/p, & \text{with probability } p \\ \boldsymbol{0}, & \text{with probability } 1 - p \end{cases}. \qquad (2)$$

$p$ determines the typical percentage of channels included, and was set as $p = 0.75$. It is straightforward to create deeper variants of the model by augmenting SyncNet with additional standard convolutional

layers. However, in our experiments, adding more layers typically resulted in over-fitting due to the limited numbers of training samples, but will likely be beneficial in larger datasets.

## 2.2 SyncNet Targets Class Differences in Cross-Spectral Densities

The cross-spectral density [3] is a widely used metric for understanding the synchronous nature of signal in frequency bands. The cross-spectral density is typically constructed by converting a time-series into a frequency representation, and then calculating the complex covariance matrix in each frequency band. In this section we sketch how the SyncNet filter bank targets cross-spectral densities to make optimal classifications. The discussion will be in the complex domain first, and then it will be demonstrated why the same result occurs in the real domain.

In the time-domain, it is possible to understand the cross-spectral density of a single frequency band by using a cross-spectral kernel [30] to define the covariance function of a Gaussian process. Letting $\tau = t - t'$, the cross-spectral kernel is defined

$$\boldsymbol{K}^{CSD}_{cc't t'} = \mathrm{cov}(x_{ct}, x_{c't'}) = A_{cc'}\kappa(\tau), \quad \kappa(\tau) = \exp\left(-\tfrac{1}{2}\beta^*\tau^2 + \jmath\omega^*\tau\right). \tag{3}$$

Here, $\omega^*$ and $\beta^*$ control the frequency band. $c$ and $c'$ are channel indexes. $\mathbf{A} \in \mathbb{C}^{C \times C}$ is a positive semi-definite matrix that defines the cross-spectral density for that frequency band controlled by $\kappa(\tau)$. Each entry $A_{cc'}$ is made of of a magnitude $|A_{cc'}|$ that controls the power (amplitude) coherence between electrodes in that frequency band and a complex phase that determines the optimal time offset between the signals. The covariance over the complete multi-channel times series is given by $\boldsymbol{K}^{CSD} = \boldsymbol{A} \otimes \kappa(\boldsymbol{\tau})$. The power (magnitude) coherence is given by the absolute value of the entry, and the phase offset can be determined by the rotation in the complex space.

A generative model for oscillatory neural signals is given by a Gaussian process with this kernel [30], where $\mathrm{vec}(\mathbf{X}) \sim \mathcal{CN}(\mathbf{0}, \boldsymbol{K}^{CSD} + \sigma^2 I_{C \times T})$. The entries of $\boldsymbol{K}^{CSD}$ are given from (3). $\mathcal{CN}$ denotes the circularly symmetric complex normal. The additive noise term $\sigma^2 I_{C \times T}$ is excluded in the following for clarity.

Note that the complex form of (1) in SyncNet across channels is given as $f(\tau) = f_\omega(\tau)\boldsymbol{s}$, where $f_\omega(\tau) = \exp(-\tfrac{1}{2}\beta\tau^2 + \jmath\omega\tau)$ is the filter over time and $\boldsymbol{s} = \boldsymbol{b} \odot \exp(\jmath\phi)$ are the weights and rotations of a single SyncNet filter. Suppose that each channel was filtered independently by the filter $\boldsymbol{f}_\omega = f_\omega(\boldsymbol{\tau})$ with a vector input $\boldsymbol{\tau}$. Writing the convolution in matrix form as $\tilde{\mathbf{x}}_c = \mathbf{f}_\omega * \mathbf{x}_c = \boldsymbol{F}^\dagger_\omega \boldsymbol{x}_c$, where $\boldsymbol{F}_\omega \in \mathbb{C}^{T \times T}$ is a matrix formulation of the convolution operator, results in a filtered signal $\tilde{\mathbf{x}}_c \sim \mathcal{CN}\left(\mathbf{0}, A_{cc}\boldsymbol{F}^\dagger_\omega\kappa(\boldsymbol{\tau})\boldsymbol{F}_\omega\right)$. For a filtered version over all channels, $\boldsymbol{X}^T = [\boldsymbol{x}^T_1, \cdots, \boldsymbol{x}^T_C]$, the distribution would be given by

$$\mathrm{vec}(\tilde{\mathbf{X}}) = \mathrm{vec}(\boldsymbol{F}^\dagger_\omega \boldsymbol{X}^T) \sim \mathcal{CN}\left(\mathbf{0}, \boldsymbol{A} \otimes \boldsymbol{F}^\dagger_\omega\kappa(\boldsymbol{\tau})\boldsymbol{F}_\omega\right), \quad \tilde{\mathbf{x}}_t \sim \mathcal{CN}(\mathbf{0}, \mathbf{A}\left[\boldsymbol{F}^\dagger_\omega\kappa(\boldsymbol{\tau})\boldsymbol{F}_\omega\right]_{tt}). \tag{4}$$

$\tilde{\mathbf{x}}_t \in \mathbb{R}^C$ is defined as the observation at time $t$ for all $C$ channels. The diagonal of $\left[\boldsymbol{F}^\dagger_\omega\kappa(\boldsymbol{\tau})\boldsymbol{F}_\omega\right]$ will reach a steady-state quickly away from the edge effects, so we state this as $\mathrm{const} = \left[\boldsymbol{F}^\dagger_\omega\kappa(\boldsymbol{\tau})\boldsymbol{F}_\omega\right]_{tt}$. The output from the SyncNet filter bank prior to the pooling stage is then given by $h_t = \boldsymbol{s}^\dagger\tilde{\boldsymbol{x}}_t \sim \mathcal{CN}(0, \mathrm{const} \times \mathbf{s}^\dagger\mathbf{A}\mathbf{s})$. We note that the signal-to-noise ratio would be maximized by matching the filter's ($\boldsymbol{f}_\omega$) frequency properties to the generated frequency properties; i.e. $\beta$ and $\omega$ from (1) should match $\beta^*$ and $\omega^*$ from (3).

We next focus on the properties of an optimal $\boldsymbol{s}$. Suppose that two classes are generated from (3) with cross-spectral densities of $\mathbf{A}_0$ and $\mathbf{A}_1$ for classes 0 and 1, respectively. Thus, the signals are drawn from $\mathcal{CN}(\mathbf{0}, \boldsymbol{A}_y \otimes \kappa(\boldsymbol{\tau}))$ for $y = \{0, 1\}$. The optimal projection $\boldsymbol{s}^*$ would maximize the differences in the distribution $h_t$ depending on the class, which is equivalent to maximizing the ratio between the variances of the two cases. Mathematically, this is equivalent to finding

$$\mathbf{s}^* = \arg\max_{\mathbf{s}} \max\left\{\frac{\mathbf{s}^\dagger\mathbf{A}_1\mathbf{s}}{\mathbf{s}^\dagger\mathbf{A}_0\mathbf{s}}, \frac{\mathbf{s}^\dagger\mathbf{A}_0\mathbf{s}}{\mathbf{s}^\dagger\mathbf{A}_1\mathbf{s}}\right\} = \arg\max_{\mathbf{s}} |\log(\mathbf{s}^\dagger\mathbf{A}_1\mathbf{s}) - \log(\mathbf{s}^\dagger\mathbf{A}_0\mathbf{s})|. \tag{5}$$

Note that the constant dropped out due to the ratio. Because the SyncNet filter is attempting to classify the two conditions, it should learn to best differentiate the classes and match the optimal $\boldsymbol{s}^*$. We demonstrate in Section 5.1 on synthetic data that SyncNet filters do in fact align with this optimal direction and is therefore targeting properties of the cross-spectral densities.

In the above discussion, the argument was made with respect to complex signals and models; however, a similar result holds when only the real domain is used. Note that if the signals are oscillatory, then

the result after the filtering of the domain and the max-pooling will be essentially the same as using a max-pooling on the absolute value of the complex filters. This is because the filtered signal is rotated through the complex domain, and will align with the real domain within the max-pooling period for standard signals. This is shown visually in Supplemental Figure 9.

## 3 Gaussian Process Adapter

A practical issue in EEG datasets is that electrode layouts are not constant, either due to inconsistent device design or electrode failure. Secondly, nearby electrodes are highly correlated and contain redundant information, so fitting parameters to all electrodes results in overfitting. These issues are addressed by developing a Gaussian Process (GP) adapter, in the spirit of [16], trained with SyncNet as shown in the left side of Figure 2. Regardless of the electrode layout, the observed signal $X$ at electrode locations $p = \{p_1, \cdots, p_C\}$ are mapped to a shared number of pseudo-inputs at locations $p^* = \{p_1^*, \cdots, p_L^*\}$ before being input to SyncNet.

In contrast to prior work, the proposed GP adapter is formulated as a multi-task GP [4] and the pseudo-input locations $p^*$ are *learned*. A GP is used to map $X \in \mathbb{R}^{C \times T}$ at locations $p$ to the pseudo-signals $X^* \in \mathbb{R}^{L \times T}$ at locations $p^*$, where $L < C$ is the number of pseudo-inputs. Distances are constructed by projecting each electrode into a 2D representation by the Azimuthal Equidistant Projection. When evaluated at a finite set of points, the multi-task GP [4] can be written as a multivariate normal

$$\text{vec}(X) \sim \mathcal{N}\left(f, \sigma^2 I_{C \times T}\right), \ f \sim \mathcal{N}\left(0, \mathbf{K}\right). \tag{6}$$

$\mathbf{K}$ is constructed by a kernel function $\mathcal{K}(\tau, c, c')$ that encodes separable relationships through time and through space. The full covariance matrix can be calculated as $\mathbf{K} = K_{pp} \otimes K_{tt}$, where $K_{p_c p_{c'}} = \alpha_1 \exp(-\alpha_2 ||p_c - p_{c'}||_1)$ and $K_{tt}$ is set to identity matrix $I_T$. $K_{pp} \in \mathbb{R}^{C \times C}$ targets the spatial relationship across channels using the exponential kernel. Note that this kernel $\mathbf{K}$ is distinct from $K^{CSD}$ used in section 2.2.

Let the pseudo-inputs locations be defined as $p_l^*$ for $l = 1, \cdots, L$. Using the GP formulation, the signal can be inferred at the $L$ pseudo-input locations from the original signal. Following [16], only the expectation of the signal is used (to facilitate fast computation), which is given by $X^* = \mathbb{E}(X^*|X) = K_{p^*p}(K_{pp} + \sigma^2 I_C)^{-1}X$. An illustration of the learned new locations is shown under $X^*$ in Figure 2. The derivation of this mathematical form and additional details on the GP adapter are included in Supplemental Section A.

The GP adapter parameters $p^*$, $\alpha_1$, $\alpha_2$ are optimized jointly with SyncNet. The input signal $X_i$ is mapped to $X_i^*$, which is then input to SyncNet. The predicted label $\hat{y}_i$ is given by $\hat{y}_i = \text{Sync}(X_i^*; \theta)$, where $\text{Sync}()$ is the prediction function of SyncNet. Given the SyncNet loss function $\sum_{i=1}^{N} \ell(\hat{y}_i, y_i) = \sum_{i=1}^{N} \ell(\text{Sync}(X_i^*; \theta), y_i)$, the overall training loss function

$$\mathcal{L} = \sum_{i=1}^{N} \ell\left(\text{Sync}(\mathbb{E}[X_i^*|X_i]; \theta), y_i\right) = \sum_{i=1}^{N} \ell\left(\text{Sync}(K_{p^*p}(K_{pp} + \sigma^2 I_C)^{-1}X_i; \theta), y_i\right), \tag{7}$$

is jointly minimized over the SyncNet parameters $\theta$ and the GP adapter parameters $\{p^*, \alpha_1, \alpha_2\}$. The GP uncertainty can be included in the loss at the expense of significantly increased optimization cost, but does not result in performance improvements to justify the increased cost [16].

## 4 Related Work

Frequency-spectrum features are widely used for processing EEG/LFP signals. Often this requires calculating synchrony- or entropy-based features within predefined frequency bands, such as [20, 5, 9, 14]. There are many hand-crafted features and classifiers for a BCI task [18]; however, in our experiments, these hand-crafted features did not perform well on long oscillatory signals. The EEG signal is modeled in [1] as a matrix-variate model with spatial and spectral smoothing. However, the number of parameters scales with time length, rendering the approach ineffective for longer time series. A range-EEG feature has been proposed [23], which measures the peak-to-peak amplitude. In contrast, our approach learns frequency bands of interest and we can deal with long time series evaluated in our experiments.

Deep learning has been a popular recent area of research in EEG analysis. This includes Restricted Boltzmann Machines and Deep Belief Networks [17, 36], CNNs [32, 29], and RNNs [2, 34]. These

approaches focus on learning both spatial and temporal relationships. In contrast to hand-crafted features and SyncNet, these deep learning methods are typically used as a black box classifier. EEGNET [15] considered a four-layer CNN to classify event-related potentials and oscillatory EEG signals, demonstrating improved performance over low-level feature extraction. This network was designed to have limited parameters, requiring 2200 for their smallest model. In contrast, the SyncNet filters are simple to interpret and require learning only a few hundred parameters.

An alternative approach is to design GP kernels to target synchrony properties and learn appropriate frequency bands. The phase/amplitude synchrony of LFP signals has been modeled [30, 10] with the cross-spectral mixture (CSM) kernel. This approach was used to define a generative model over differing classes and may be used to learn an unsupervised clustering model. A key issue with the CSM approach is the computational complexity, where gradients cost $\mathcal{O}(NTC^3)$ (using approximations), and is infeasible with the larger number of electrodes in EEG data. In contrast, the proposed GP adapter requires only a single matrix inversion shared by most data points, which is $\mathcal{O}(C^3)$.

The use of wavelets has previously been considered in scattering networks [6]. Scattering networks used Morlet wavelets for image classification, but did not consider the complex rotation of wavelets over channels nor the learning of the wavelet widths and frequencies considered here.

## 5  Experiments

To demonstrate that SyncNet is targeting synchrony information, we first apply it to synthetic data in Section 5.1. Notably, the learned filter bank recovers the optimal separating filter. Empirical performance is given for several EEG datasets in Section 5.2, where SyncNet often has the highest hold-out accuracy while maintaining interpretable features. The usefulness of the GP adapter to combine datasets is demonstrated in Section 5.3, where classification performance is dramatically improved via data augmentation. Empirical performance on an LFP dataset is shown in Section 5.4. Both the LFP signals and the EEG signals measure broad voltage fluctuations from the brain, but the LFP has a significantly cleaner signal because it is measured inside the cortical tissue. In all tested cases, SyncNet methods have essentially state-of-the-art prediction while maintaining interpretable features.

The code is written in Python and Tensorflow. The experiments were run on a 6-core i7 machine with a Nvidia Titan X Pascal GPU. Details on training are given in Supplemental Section C.

### 5.1  Synthetic Dataset

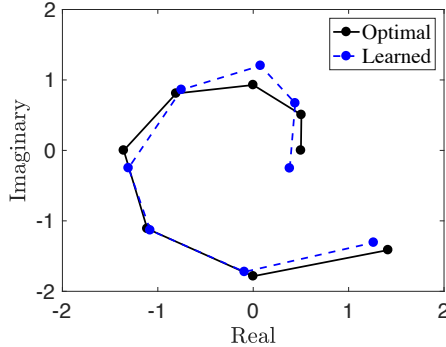

Synthetic data are generated for two classes by drawing data from a circularly symmetric normal matching the synchrony assumptions discussed in Section 2.2. The frequency band is pre-defined as $\omega^* = 10Hz$ and $\beta^*$ is defined as 40 (frequency variance of $2.5Hz$) in (3). The number of channels is set to $C = 8$. Example data generated by this procedure is shown in Figure 1 (Right), where only the real part of the signal is kept.

Figure 3: Each dot represents one of $8$ electrodes. The dots give complex directions for optimal and learned filters, demonstrating that SyncNet approximately recovers optimal filters.

$A_1$ and $A_0$ are set such that the optimal vector from solving (5) is given by the shape visualized in Figure 3. This is accomplished by setting $A_0 = I_C$ and $A_1 = I + s^*(s^*)^\dagger$. Data is then simulated by drawing from $\text{vec}(\mathbf{X}) \sim \mathcal{CN}(\mathbf{0}, K^{CSD} + \sigma^2 I_{C \times T})$ and keeping only the real part of the signal. $K^{CSD}$ is defined in equation (3) with $\mathbf{A}$ set to $A_0$ or $A_1$ depending on the class. In this experiment, the goal is to relate the filter learned in SyncNet and to this optimal separating plane $s^*$.

To show that SyncNet is targeting synchrony, it is trained on this synthetic data using only one single convolutional filter. The learned filter parameters are projected to the complex space by $s = b \odot \exp(\jmath\phi)$, and are shown overlaid (rotated and rescaled to handle degeneracies) with the

optimal rotations in Figure 3. As the amount of data increases, the SyncNet filter recovers the expected relationship between channels and the predefined frequency band. In addition, the learned $\omega$ is centered at $11Hz$, which is close to the generated feature band $\omega^*$ of $10Hz$. These synthetic data results demonstrate that SyncNet is able to recover frequency bands of interest and target synchrony properties.

## 5.2 Performance on EEG Datasets

We consider three publicly available datasets for EEG classification, described below. After the validation on the publicly available data, we then apply the method to a new clinical-trial data, to demonstrate that the approach can learn interpretable features that track the brain dynamics as a result of treatment.

**UCI EEG**: This dataset[2] has a total of 122 subjects with 77 diagnosed with alcoholism and 45 control subjects. Each subject undergoes 120 separate trials. The stimuli are pictures selected from 1980 Snodgrass and Vanderwart picture set. The EEG signal is of length one second and is sampled at 256Hz with 64 electrodes. We evaluate the data both within subject, which is randomly split as $7:1:2$ for training, validation and testing, and using 11 subjects rotating test set. The classification task is to recover whether the subject has been diagnosed with alcoholism or is a control subject.

**DEAP dataset**: The "Database for Emotion Analysis using Physiological signals" [14] has a total of 32 participants. Each subject has EEG recorded from 32 electrodes while they are shown a total of 40 one-minute long music videos with strong emotional score. After watching each video, each subject gave an integer score from one to nine to evaluate their feelings in four different categories. The self-assessment standards are valence (happy/unhappy), arousal (bored/excited), dominance (submissive/empowered) and personal liking of the video. Following [14], this is treated as a binary classification with a threshold at a score of 4.5. The performance is evaluated with leave-one-out testing, and the remaining subjects are split to use 22 for training and 9 for validation.

**SEED dataset**: This dataset [35] involves repeated tests on 15 subjects. Each subject watches 15 movie clips 3 times. It clip is designated with a negative/neutral/positive emotion label, while the EEG signal is recorded at $1000Hz$ from 62 electrodes. For this dataset, leave-one-out cross-validation is used, and the remaining 14 subjects are split with 10 for training and 4 for validation.

**ASD dataset**: The Autism Spectral Disorder (ASD) dataset involves 22 children from ages 3 to 7 years undergoing treatment for ASD with EEG measurements at baseline, 6 months post treatment, and 12 months post treatment. Each recording session involves 3 one-minute videos designed to measure responses to social stimuli and controls, measured with a 121 electrode array. The trial was approved by the Duke Hospital Institutional Review Board and conducted under IND #15949. Full details on the experiments and initial clinical results are available [7]. The classification task is to predict the time relative to treatment to track the change in neural signatures post-treatment. The cross-patient predictive ability is estimated with leave-one-out cross-validation, where 17 patients are used to train the model and 4 patients are used as a validation set.

| Dataset | UCI | | DEAP [14] | | | | SEED [35] | ASD |
|---|---|---|---|---|---|---|---|---|
| | Within | Cross | Arousal | Valence | Domin. | Liking | Emotion | Stage |
| DE [35] | 0.821 | 0.622 | 0.529 | 0.517 | 0.528 | 0.577 | 0.491 | 0.504 |
| PSD [35] | 0.816 | 0.605 | 0.584 | 0.559 | 0.595 | 0.644 | 0.352 | 0.499 |
| rEEG [23] | 0.702 | 0.614 | 0.549 | 0.538 | 0.557 | 0.585 | 0.468 | 0.361 |
| Spectral [14] | * | * | **0.620** | 0.576 | * | 0.554 | * | * |
| EEGNET [15] | 0.878 | 0.672 | 0.536 | 0.572 | 0.589 | 0.594 | 0.533 | 0.363 |
| MC-DCNN [37] | 0.840 | 0.300 | 0.593 | 0.604 | 0.635 | 0.621 | 0.527 | 0.584 |
| SyncNet | 0.918 | 0.705 | 0.611 | 0.608 | **0.651** | **0.679** | **0.558** | 0.630 |
| GP-SyncNet | **0.923** | **0.723** | 0.592 | **0.611** | 0.621 | 0.659 | 0.516 | **0.637** |

Table 1: Classification accuracy on EEG datasets.

The accuracy of predictions on these EEG datasets, from a variety of methods, is given in Table 1. We also implemented other hand-crafted spatial features, such as the brain symmetric index [31]; however, their performance was not competitive with the results here. EEGNET is an EEG-specific convolutional network proposed in [15]. The "Spectral" method from [14] uses an SVM on extracted

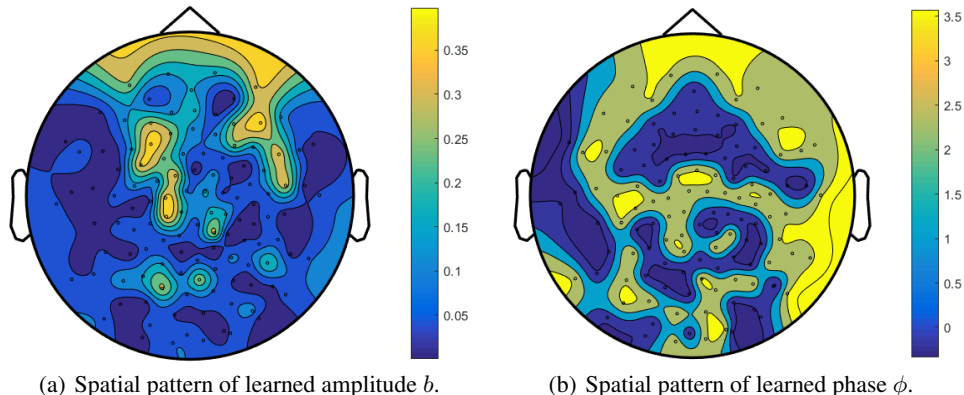

| (a) Spatial pattern of learned amplitude $b$. | (b) Spatial pattern of learned phase $\phi$. |

Figure 4: Learned filter centered at 14Hz on the ASD dataset. Figures made with FieldTrip [22].

spectral power features from each electrode in different frequency bands. MC-DCNN [37] denotes a 1D CNN where the filters are learned without the constraints of the parameterized structure. The SyncNet used 10 filter sets both with (GP-SyncNet) and without the GP adapter. Remarkably, the basic SyncNet already delivers state-of-the-art performance on most tasks. In contrast, the hand-crafted features did not effectively cannot capture available information and the alternative CNN based methods severely overfit the training data due to the large number of free parameters.

In addition to state-of-the-art classification performance, a key component of SyncNet is that the features extracted and used in the classification are *interpretable*. Specifically, on the ASD dataset, the proposed method significantly improves the state-of-the-art. However, the end goal of this experiment is to understand how the neural activity is changing in response to the treatment. On this task, the ability of SyncNet to visualize features is important for dissemination to medical practitioners. To demonstrate how the filters can be visualized and communicated, we show one of the filters learned in SyncNet on the ASD dataset in Figure 4. This filter, centered at 14Hz, is highly associated with the session at 6 months post-treatment. Notably, this filter bank is dominantly using the signals measured at the forward part of the scalp (Figure 4, Left). Intriguingly, the phase relationships are primarily in phase for the frontal regions, but note that there are off-phase relationships between the midfrontal and the frontal part of the scale (Figure 4, Right). Additional visualizations of the results are given in Supplemental Section E.

### 5.3 Experiments on GP adapter

In the previous section, it was noted that the GP adapter can improve performance within an existing dataset, demonstrating that the GP adapter is useful to reduce the number of parameters. However, our primary designed use of the GP Adapter is to unify different electrode layouts. This is explored further by applying the GP-SyncNet to the UCI EEG dataset and changing the number of pseudo-inputs. Notably, a mild reduction in the number of pseudo-inputs improves performance over directly using the measured data (Supplemental Figure 6(a)) by reducing the total number of parameters. This is especially true when comparing the GP adapter to using a random subset of channels to reduce dimensionality.

|  | SyncNet | GP-SyncNet | GP-SyncNet Joint |
|---|---|---|---|
| DEAP [14] dataset | $0.521 \pm 0.026$ | $0.557 \pm 0.025$ | $\mathbf{0.603} \pm 0.020$ |
| SEED [35] dataset | $0.771 \pm 0.009$ | $0.762 \pm 0.015$ | $\mathbf{0.779} \pm 0.009$ |

Table 2: Accuracy mean and standard errors for training two datasets separately and jointly.

To demonstrate that the GP adapter can be used to combine datasets, the DEAP and SEED datasets were trained jointly using a GP adapter. The SEED data was downsampled to $128Hz$ to match the frequency of DEAP dataset, and the data was separated into $4$ second windows due to their different lengths. The label for the trial is attached for each window. To combine the labeling space, only the negative and positive emotion labels were kept in SEED and valence was used in the DEAP dataset. The number of pseudo-inputs is set to $L = 26$. The results are given in Table 2, which demonstrates that combining datasets can lead to dramatically improved generalization ability due to the data

augmentation. Note that the basic SyncNet performances in Table 2 differ from the results in Table 1. Specifically, the DEAP dataset performance is worse; this is due to significantly reduced information when considering a 4 second window instead of a 60 second window. Second, the performance on SEED has improved; this is due to considering only 2 classes instead of 3.

## 5.4 Performance on an LFP Dataset

Due to the limited publicly available multi-region LFP datasets, only a single LFP data was included in the experiments. The intention of this experiment is to show that the method is broadly applicable in neural measurements, and will be useful with the increasing availability of multi-region datasets. An LFP dataset is recorded from 26 mice from two genetic backgrounds (14 wild-type and 12 CLOCK$\Delta$19). CLOCK$\Delta$19 mice are an animal model of a psychiatric disorder. The data are sampled at 200 Hz for 11 channels. The data recording from each mouse has five minutes in its home cage, five minutes from an open field test, and ten minutes from a tail-suspension test. The data are split into temporal windows of five seconds. SyncNet is evaluated by two distinct prediction tasks. The first task is to predict the genotype (wild-type or CLOCK$\Delta$19) and the second task is to predict the current behavior condition (home cage, open field, or tail-suspension test). We separate the data randomly as $7 : 1 : 2$ for training, validation and testing

|  | PCA + SVM | DE [35] | PSD [35] | rEEG [23] | EEGNET [15] | SyncNet |
|---|---|---|---|---|---|---|
| Behavior | 0.911 | 0.874 | 0.858 | 0.353 | 0.439 | **0.946** |
| Genotype | 0.724 | 0.771 | 0.761 | 0.449 | 0.689 | **0.926** |

Table 3: Comparison between different methods on an LFP dataset.

Results from these two predictive tasks are shown in Table 3. SyncNet used $K = 20$ filters with filter length $40$. These results demonstrate that SyncNet straightforwardly adapts to both EEG and LFP data. These data will be released with publication of the paper.

## 6 Conclusion

We have proposed SyncNet, a new framework for EEG and LFP data classification that learns interpretable features. In addition to our original architecture, we have proposed a GP adapter to unify electrode layouts. Experimental results on both LFP and EEG data show that SyncNet outperforms conventional CNN architectures and all compared classification approaches. Importantly, the features from SyncNet can be clearly visualized and described, allowing them to be used to understand the dynamics of neural activity.

## Acknowledgements

In working on this project L.C. received funding from the DARPA HIST program; K.D., L.C., and D.C. received funding from the National Institutes of Health by grant R01MH099192-05S2; K.D received funding from the W.M. Keck Foundation; G.D. received funding from Marcus Foundation, Perkin Elmer, Stylli Translational Neuroscience Award, and NICHD 1P50HD093074.

## Footnotes

[1]It is straightforward to use the Morlet wavelet directly and define the outputs as complex variables and define the neural network to target the same properties, but this leads to both computational and coding overhead.

[2] https://kdd.ics.uci.edu/databases/eeg/eeg.html

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
