[Supplementary Material]

# Supplemental Materials:

## A  Additional Description of the GP-SyncNet

As was included in the main paper, $\boldsymbol{p}$ denotes a location of a real electrode locations and $\boldsymbol{p}^*$ is a pseudo-input locations. $\boldsymbol{X} \in \mathbb{R}^{C \times T}$ is the real electrode signals and $\boldsymbol{X}^* \in \mathbb{R}^{L \times T}$ is the signal at pseudo-input locations. Assume there are $C$ electrodes $\boldsymbol{p} = \{\boldsymbol{p}_c\}_{c=1,\cdots,C}$ and $L$ pseudo-input locations $\boldsymbol{p}_l^*$, $l = 1, \cdots, L$. For the original signal $\boldsymbol{X}$, the distribution is given in (6).

The signal $\boldsymbol{X}^*$ is also given by Gaussian process properties. The covariances between the locations $\{\boldsymbol{p}_c\}_{1,\ldots,C}$ and $\{\boldsymbol{p}_\ell^*\}_{\ell=1,\ldots,}$ are constructed in the matrix $\boldsymbol{K}_{\boldsymbol{p}^*\boldsymbol{p}} \in \mathbb{R}^{L \times C}$. Then, $p(\text{vec}(\boldsymbol{X}^*)|\boldsymbol{X})$ is given by a normal distribution $\mathcal{N}(\boldsymbol{\mu}, \boldsymbol{\Sigma})$ with,

$$\boldsymbol{\mu} = \text{vec}\left(\boldsymbol{K}_{\boldsymbol{p}^*\boldsymbol{p}}(\boldsymbol{K}_{\boldsymbol{pp}} + \sigma^2 \boldsymbol{I}_C)^{-1}\boldsymbol{X}\right), \boldsymbol{\Sigma} = \left(\boldsymbol{K}_{\boldsymbol{p}^*\boldsymbol{p}^*} - \boldsymbol{K}_{\boldsymbol{p}^*\boldsymbol{p}}(\boldsymbol{K}_{\boldsymbol{pp}} + \sigma^2 \boldsymbol{I}_C)^{-1}\boldsymbol{K}_{\boldsymbol{pp}^*}\right) \otimes \boldsymbol{I}_T.$$

Ignoring the variance, the pseudo-input signal is given as $\mathbb{E}(\boldsymbol{X}^*|\boldsymbol{X}) = \boldsymbol{K}_{\boldsymbol{p}^*\boldsymbol{p}}(\boldsymbol{K}_{\boldsymbol{pp}} + \sigma^2 \boldsymbol{I}_C)^{-1}\boldsymbol{X}$. This form depends on the relationship that

$$\left(\boldsymbol{K}_{\boldsymbol{p}^*\boldsymbol{p}}(\boldsymbol{K}_{\boldsymbol{pp}} + \sigma^2 \boldsymbol{I}_C)^{-1}\right) \otimes \boldsymbol{I}_T \text{vec}(\boldsymbol{X}_i) = \boldsymbol{K}_{\boldsymbol{p}^*\boldsymbol{p}}(\boldsymbol{K}_{\boldsymbol{pp}} + \sigma^2 \boldsymbol{I}_C)^{-1}\boldsymbol{X}_i.$$

Note that the pseudo-input locations $\boldsymbol{p}^*$ are learned during model training. Fast matrix inversions can be performed for inference because $C$ is relatively small and $T$ has only linear computational dependence due to the Kronecker structure. In contrast, [16] used approximations to the inverse covariance matrix to facilitate fast inference; such approximations are not necessary here and they would limit the locations of the pseudo-inputs. Since the loss is a deterministic function of $\boldsymbol{X}_i$, the derivative with respect to parameters in GP adapter and SyncNet can be derived by backprop. The pseudo-inputs both unify the location information and reduce free parameter number by compressing the channel number.

If desired, the uncertainty of the signal can be included in the network using equation (8). $\ell$ is a loss function, which is cross-entropy loss for SyncNet. Sync is the prediction function of SyncNet. $\mu_i$ and $\Sigma_i$ are given in equation (8).

$$\boldsymbol{\eta}^*, \boldsymbol{\theta}^* = \arg\min_{\boldsymbol{\eta}, \boldsymbol{\theta}} \sum_{i=1}^N \mathbb{E}_{\boldsymbol{X}_i^* \sim \mathcal{N}(\boldsymbol{\mu}_i, \boldsymbol{\Sigma}_i)}\left[\ell\left(\text{Sync}(\boldsymbol{X}_i^*; \boldsymbol{\theta}), y_i\right)\right] \tag{8}$$

However, [16] demonstrated that considering the uncertainty gives only a small gain in performance but high increase in computational complexity. Our own experiments validated their conclusion.

## B  Autoencoded SyncNet (AE-SyncNet)

An autoencoder is typically used to facilitate *unsupervised* deep learning methods, as the autoencoder is used to learn features that may also be used to accurately reconstruct the original signal. However, it has been shown that including reconstruction information via an autoencoder can improve predictive performance even in a fully supervised task [27]. In addition, autoencoder structures have proven effective in semi-supervised learning, where there is an abundance of unlabeled data and comparatively few labels [27, 25]. Therefore, an autoencoder version of the SyncNet (AE-SyncNet) is developed both to address overfitting in the supervised case and to facilitate semi-supervised learning. We give a graphical description of the autoencoder combined with the GP Adapter in Figure 5.

The autoencoder shares the same representation of the SyncNet up to the filter output $\boldsymbol{h}^{(k)}$ given in (1) and (9).

$$\boldsymbol{h}^{(k)} = \sum_{c=1}^C \boldsymbol{f}_c^{(k)} * \boldsymbol{x}_c, \ k = 1, \cdots, K. \tag{9}$$

At this point the network bifurcates, and these filters feed to the reconstruction network and the classification network. In the reconstruction network, a separate max-pooling occurs over a much smaller temporal band (two in our experiments), and its outputs are immediately unpooled from $\tilde{\boldsymbol{h}}^{(k)}$ to recover the original dimensionality and is denoted $\hat{\boldsymbol{h}}^{(k)}$, where the entries of $\hat{\boldsymbol{h}}^{(k)}$ in each pooling region are set to the maximum value of that region in $\boldsymbol{h}^{(k)}$.

Figure 5: The framework of GP-AE-SyncNet. There is a GP adapter on both the encoder and decoder, which learns new locations and use GP to project signals to and from the pseudo-inputs. The plot below input $X$ is standard electrode layout. The plot below $\hat{X}^*$ is the learned signal at anchor positions. $\hat{X}$ is the reconstruction.

The original signals are then reconstructed by convolving the $\hat{h}^{(k)}$ for $k = 1, \ldots, K$ with a set of "deconvolutional" filters. Each deconvolutional filter is designed to mirror the SyncNet filters in (1) and are defined as

$$f_c'^{(k)}(\tau) = b_c'^{(k)} \cos(\omega'^{(k)}\tau + \phi'^{(k)}_c) \exp(-\beta'^{(k)}\tau^2).$$

The $'$ denotes a separation parameterization (i.e. $b \neq b'$) for the individual parameters in the autoencoder structure to emphasize that the parameters are not tied together as they would be in a wavelet reconstruction. The final reconstruction is defined as

$$\hat{x}_c = \sum_{k=1}^{K} f_c'^{(k)} * \hat{h}^{(k)}. \tag{10}$$

When training the model, the reconstruction loss is simply defined as the squared Frobenius norm between the original section and the reconstruction, $\ell_{AR}(\hat{X}, X) = ||\hat{X}_i - X||_F^2$. When training a semi-supervised framework, the relative strengths of the reconstruction loss and the classification loss $\ell_{class}(g(X_i), y_i)$ is chosen to maximize predictive performance. Therefore, there is a tuning parameter $\gamma$ to modulate the total loss per data point $\ell_{AR}(\hat{X}_i, X_i) + \gamma \ell_{class}(g(X_i; \theta), y_i)$. If the model is trained in a semi-supervised manner, $\ell_{class}$ is ignored whenever a data example does not have a known associated label.

### B.1  Including the GP Adapter (GP-AE-SyncNet)

The GP Adapter can be straightforwardly included with the autoencoder structure. This combination is referred to as the GP-AE-SyncNet. In the decoder, the GP Adapter must project from the pseudo-input locations $p^*$ back to the original input points.

The mean of this projection results in a similar form to (6), where

$$\hat{X}_i = K_{pp^*}(K_{p^*p^*} + \sigma^2 I_{C'})^{-1} \hat{X}_i^*.$$

The framework of GP-AE-SyncNet is given in Figure 5. In the GP-AE-SyncNet, the final reconstruction loss is on the difference between the reconstructed $\hat{X}_i$ and the original $X_i$.

## C  Model Training

The model is trained by stochastic gradient methods using the Adam optimization algorithm [13]. The GP adapter parameters $\eta = \{p^*, \alpha_1, \alpha_2\}$, and the SyncNet parameters $\theta = \{\omega, b, \phi, \beta, \ldots\}$ are learned. The "..." denotes additional parameterizations for deeper structures, if desired.

Figure 6: (a) Evaluation of the GP adapter on AUC for different $\frac{|\boldsymbol{p}^*|}{|\boldsymbol{p}|}$ ratios on the UCI EEG dataset. The classification performance peaks at a mild reduction in the number of inputs, and the GP is more effective than randomly keeping channels. (b) The AE-SyncNet improves over the purely supervised solution when only labeling a fraction of the data points.

The training of GP-SyncNet can be performed by first fixing parameters $\boldsymbol{\eta}$ in the GP to a reasonable initialization, while $\boldsymbol{\theta}$ is updated. After this pretraining, $\boldsymbol{\eta}$ and $\boldsymbol{\theta}$ are jointly updated. This prevents the parameters from falling into a bad local optimum. We initialize the frequency parameters $\omega$ in eq.1 with random uniform samples in the range of $[0, 50 * 2\pi\Delta]$, where $\Delta^{-1}$ is the signal sampling rate, setting a practical range of 0-50 $Hz$ for the frequency bands. The amplitude $\boldsymbol{b}$ is i.i.d sampled from $\mathcal{N}(0, 0.01)$. All the phase shifts are initially set to zero. To create a reasonable initialization for the GP adapter, $\boldsymbol{p}^*$ is initialized at randomly selected channel locations, and $\alpha_1$ and $\alpha_2$ in the GP adapter are initialized as 1.0 and 2.0, respectively.

# D  Additional Experiments on GP-SyncNet and AE-SyncNet

## D.1  Experiment on Gaussian Process Adapter

As mentioned in Section 3, the GP adapter smooths spatial information and reduces the number of free parameters. This is evaluated on the UCI EEG dataset within patients. In Figure 6(a), the classification AUC for different number of pseudo-inputs and randomly keep original channels is shown.

When the ratio of channels $\frac{|\boldsymbol{p}^*|}{|\boldsymbol{p}|}$ is below 0.7, increasing the new location number will improve the performance of the SyncNet. However, at some point introducing more new locations $\boldsymbol{p}^*$ causes over fitting. The GP adapter can achieve a mild improvement by decreasing the dimensionality. Note that $\frac{|\boldsymbol{p}^*|}{|\boldsymbol{p}|}$ can also be larger than one, but this is not empirically beneficial.

## D.2  Autoencoded SyncNet (AE-SyncNet)

Figure 6(b) shows the effect of reducing the number of supervised points on UCI dataset. The training and testing samples are within and cross subjects, which is in correspondence to the first column of Table 1. Note that the autoencoder structure leverages unlabeled points in order to improve the classification prediction (semi-supervised learning). This gives some moderate improvements.

| Dataset | UCI | DEAP [14] | | | | LFP | |
|---|---|---|---|---|---|---|---|
| | Within | Arousal | Valence | Domin. | Liking | Behavior | Genotype |
| AE-SyncNet | 0.921 | 0.620 | 0.611 | 0.647 | 0.675 | 0.959 | 0.945 |
| GP-AE-SyncNet | 0.931 | 0.608 | 0.570 | 0.624 | 0.652 | * | * |

Table 4: Classification accuracy on EEG datasets for AE-SyncNet and GP-AE-SyncNet.

# E  Additional Visualizations of ASD Results

Here we provide additional visualizations of the learned features in the ASD dataset. In Figure 7, we give the average hidden weight $\tilde{h}_k$ for each subject at each trial point (0, 6, and 12 months) over

$K = 6$ learned filter sets. Stage $T1$ is the baseline result immediately previous to treatment. $T2$ and $T3$ are six and twelve months after the treatment, respectively. Because not all subjects were tested at all stages, there are different numbers of patients at each time point. Note that temporal dynamics on the filter weights can clearly be seen. For example, the average weight on filter 2 becomes stronger from $T1$ to $T3$, while filter 5 shows an opposite trend.

The amplitude and phase shift for these two filters are given in Figure 8. As is demonstrated in Figure 8(a), this high frequency filter increases in importance during the later stages of the trial. In contrast, Figure 8(c) shows that the relatively low frequency signal becomes less dominant after the treatment.

Figure 7: Three stages filter weight comparison. As the treatment goes on, we can see changes in the brain regions.

(a) Amplitude for filter at $42$Hz. (Index 2)

(b) Phase for filter at $42$Hz. (Index 2)

(c) Amplitude for filter at $29$Hz. (Index 5)

(d) Phase for filter at $29$Hz. (Index 5)

Figure 8: Visualizations of additional filters learned from the ASD dataset. The indexes match the assignments in Figure 7.

# F   Additional Supplemental Figures

The robustness of the SyncNet filter to using real projections rather than complex projections is shown in Figure 9. Note that after the max pool step, there is essentially no difference between the real and complex magnitude signals.

The electrode layouts used to combine experiments are shown in Figure 10.

Figure 9: The solid black line shows $h^{(k)}$ on a 8-channel signal compared to using Morlet wavelets and magnitudes in a dashed blue line. Max-pooling $h^{(k)}$ over time approximates the magnitude and reduces temporal redundancy.

(a)

(b)

Figure 10: (a) The 32-electrodes layout for DEAP dataset [14]. (b) The 62-electrodes layout for SEED dataset [35]. In section 5.3, the GP adapter is used to training on the two datasets jointly.