[Reviews · NeurIPS 2017]

Reviewer 1



Dear authors, I am not familiar with the methods you present in this paper and the methodological part is beyond my understanding. Therefore, I cannot say much about your paper apart from: - The problem you pose is interesting. Extracting synchrony features is not an easy problem. However, I am not familiar with DNNs. I am not able to evaluate whether CNN or another type should be used. - It is interesting that you checked for features used in other related fields like BCIs. You say that he recordings you use are too long. How long are they? For LFP you seem to use 5 seconds, this is not too different from BCI features. - Could the GP adapter work separately from the NN? It would be interesting in general if it could help reducing the number of electrodes when the analyses should be done in pairs of sensors. In general, the paper is very dense. Maybe this is because it is hard to follow for me.

Reviewer 2



The manuscript introduces the novel CNN named SyncNet, which was designed to detect synchrony/spectral coherence in brain electrophysiological signals. SyncNet uses parameterized convolutional filters. To overcome common issues in EEG (varying setup configuration, correlated channels, ...) a Gaussian Process Adaptor is used to transform EEG data into a pseudo input space. The design of SyncNet is chosen to allow interpretation of learned featruers. The method was applied to several publicly available datasets and its performance compared to state of the art algorithms. Results suggest that SyncNet yields competitive results. The topic of the paper is timely and important and the presented approach very interesting. The paper is well organized and mostly clear, and the reviewer agrees with the authors reasoning. There are, however, some issues that need addressing: - Are the computed SyncNet results significantly better than the results obtained with state of the art methods? - One key feature of SynchNet is interpretability of features. Are the results summarized in Fig.4 meaningful? - LFPs are only mentioned in the title and in lines 319-330. Please provide more details on LFP data and analysis or remove LFPs from the manuscript! Currently, LFP does not contribute to enhancing clarity of the paper.

Reviewer 3



This contribution introduces SyncNet, a framework combining surrogate electrode positioning via GP adaptater and convolution network with contraint convolution parameters. SyncNet is evaluated on synthetic dataset and on real datasets, with both EEG and LFP data. It compares very favorably with state-of-the-art approaches and the GP adapter plays an important role. The results are interpretable, which is of prime importance.